# Unique SMYD5 Structure Revealed by AlphaFold Correlates with Its Functional Divergence

**DOI:** 10.3390/biom12060783

**Published:** 2022-06-03

**Authors:** Yingxue Zhang, Eid Alshammari, Jacob Sobota, Alexander Yang, Chunying Li, Zhe Yang

**Affiliations:** 1Department of Biochemistry, Microbiology and Immunology, Wayne State University School of Medicine, 540 East Canfield Street, Detroit, MI 48201, USA; yingxue@wayne.edu (Y.Z.); eid.alshammari@wayne.edu (E.A.); fw1648@wayne.edu (J.S.); alexluyang22@gmail.com (A.Y.); 2Center for Molecular and Translational Medicine, Georgia State University, Atlanta, GA 30303, USA; cli19@gsu.edu

**Keywords:** SMYD5, SET and MYND domain-containing protein, structure-and-function relationships, AlphaFold, nuclear localization signal, subcellular localization

## Abstract

SMYD5 belongs to a special class of protein lysine methyltransferases with an MYND (Myeloid-Nervy-DEAF1) domain inserted into a SET (Suppressor of variegation, Enhancer of Zeste, Trithorax) domain. Despite recent advances in its functional characterization, the lack of the crystal structure has hindered our understanding of the structure-and-function relationships of this most unique member of the SMYD protein family. Here, we demonstrate the reliability of using AlphaFold structures for understanding the structure and function of SMYD5 by comparing the AlphaFold structures to the known crystal structures of SMYD proteins, using an inter-residue distance maps-based metric. We found that the AlphaFold confidence scores are inversely associated with the refined B-factors and can serve as a structural indicator of conformational flexibility. We also found that the N-terminal sequence of SMYD5, predicted to be a mitochondrial targeting signal, contains a novel non-classical nuclear localization signal. This sequence is structurally flexible and does not have a well-defined conformation, which might facilitate its recognition for SMYD5’s cytonuclear transport. The structure of SMYD5 is unique in many aspects. The “crab”-like structure with a large negatively charged cleft provides a potential binding site for basic molecules such as protamines. The less positively charged MYND domain is associated with the undetectable DNA-binding ability. The most surprising feature is an incomplete target lysine access channel that lacks the evolutionarily conserved tri-aromatic arrangement, being associated with the low H3/H4 catalytic activity. This study expands our understanding of the SMYD protein family from a classical two-lobed structure to a structure of its own kind, being as a fundamental determinant of its functional divergence.

## 1. Introduction

SET (Suppressor of variegation, Enhancer of Zeste, Trithorax) and MYND (Myeloid-Nervy-DEAF1) domain-containing proteins (SMYD) are a special class of protein lysine methyltransferases with an MYND domain inserted into a SET domain [1]. Among the five SMYD members, SMYD5 is the most unique in terms of function and structure. All SMYD members except for SMYD5 have been linked to heart and skeletal muscle development. SMYD1 is required for cardiomyocyte differentiation and the development of the right ventricle [2]. SMYD2, despite no heart phenotype upon knockout in mice, regulates sarcomere degradation under oxidative stress [3,4]. SMYD3 is essential for pericardial development in zebrafish [5], while SMYD4 is involved in eclosion in fruit flies through regulating abdominal muscle development [6]. In contrast, mice with SMYD5 knockout do not show any significant phenotypes in their heart and skeletal muscle [7]. One explanation for this difference is that SMYD5 has less prominent expression in the heart and skeletal muscle than other SMYD proteins [8]. Another argument is that such a functional divergence could be in part caused by its multiple unique structural features. SMYD5 lacks the conserved tetratricopeptide repeat (TPR)-like C-terminal domain (CTD), a helical bundle that is present in all other SMYD proteins. Since the CTD domain is involved in the formation of the substrate binding site [9], this structural difference suggests that SMYD5 might use a different mechanism in the recognition of its substrates. Another unique feature is that SMYD5 contains an extra N-terminal sequence, predicted as a mitochondrial targeting sequence, apparently being in agreement with a study showing that SMYD5 is localized to the mitochondria [10]. SMYD5 also contains two large insertions in the conserved SET and MYND domains, which are not found in other SMYD proteins. It remains unknown how these unique structural features are related to its functional divergence from other SMYD members.

Despite being still not well studied, the understanding of the biological functions of SMYD5 has been steadily increasing in the past decade, including its role in toll-like receptor 4 (TLR4) signaling in macrophages [11], maintaining chromosome integrity during embryonic stem cell differentiation [12], a potential role in metastatic breast cancer dormancy [13], and a very recent study suggesting a role in sperm chromatin remodeling via direct binding to protamines [14]. However, the lack of the crystal structure has hindered our understanding of the structure-and-function relationships of this most unique member of the SMYD protein family. A large number of crystal structures are available for SMYD2 and SMYD3, which have defined the structure of the SMYD protein family as being bilobal, with the substrate binding site located at the bottom of a cleft formed between the two lobes [15,16,17]. However, the lack of the CTD domain, which is the sole constituent of the C-lobe in other SMYD proteins, points out that the structure of SMYD5 is presumably going to be very different from other SMYD proteins. As a result of this difference, the current structures of SMYD proteins are basically deemed as not being well suited to be used as homology models to understand the relationships between SMYD5 structure and function, especially to those structural features unique to SMYD5.

In this study, we demonstrate the reliability of the use of AlphaFold (AF) structures for SMYD5 structural analysis. AF structures, predicted by a novel deep learning algorithm, have recently been shown to have more than 90% accuracy with respect to true structures, hence representing a revolutionary breakthrough that is shifting the research paradigm in protein structural analysis [18]. However, the validity of using AF structures for studying protein structure-and-function relationships has not yet been systematically examined. In this study, we carried out rigorous validation of the accuracy of AF structures by comparing the AF structures to the known crystal structures of SMYD proteins, using an RMSD-based metric and a metric we developed based on inter-residue distance maps (IRDMs). Our rigorous structural validation proves that the differences between the AF structures and the crystal structures are no larger than the differences between the crystal structures. We reveal that the unique SMYD5 structure correlates with its functional divergence in DNA binding and the histone H3/H4 catalytic activity. We also found that the N-terminal sequence, predicted as a mitochondrial targeting signal, is actually a novel non-classical nuclear localization signal, which is structurally flexible. This study expands our structural understanding of the SMYD protein family from a classical two-lobed structure to a structure of its own kind, being a fundamental determinant of its functional divergence. The approach developed for the rigorous structural validation of AF structures also provides a broadly applicable framework that can be applied to the structure-and-function analysis of any protein families with no available crystal structures.

## 2. Results

### 2.1. Comparison of AlphaFold and Crystal Structures of SMYD Proteins

AlphaFold (AF) is a deep learning algorithm that has recently been proven to have high accuracy in de novo protein structure prediction [18]. To assess the reliability of using AF structures for understanding the structure-and-function relationships of SMYD5, the accuracy of AF structures is validated by comparing these structures to the known crystal structures of SMYD proteins. For SMYD1, the only available structure is one determined by using the mouse version of the protein [15]. For SMYD2, there are twenty-two structures of the human protein and two structures of the mouse protein in the Protein Data Bank (PDB). For SMYD3, there are twenty-five known structures, all of which are of human origin. To facilitate statistical analysis, only human SMYD2 and human SMYD3 with a large number of known structures were used to evaluate the accuracy of AF structures (Appendix A). The similarities between the AF and crystal structures were evaluated by root-mean-square-deviation (RMSD) and a metric we developed based on inter-residue distance maps (IRDMs). To assess what level of structural details can be trusted for structural analysis, the comparison was performed with the C_α_ atom only, main chain atoms, or all atoms. Overall, the AF structures are not more different from the crystal structures than the differences between the crystal structures. The median RMSD between the AF structures and crystal structures is 0.46 Å for SMYD2 and 0.36 Å for SMYD3 in the C_α_ atom comparison (Figure 1A). The RMSD values calculated between the different crystal structures are comparable, with a median of 0.68 Å and 0.58 Å for SMYD2 and SMYD3, respectively. As shown by Welch’s *t*-test, the average RMSD values between the AF structures and crystal structures are statistically lower than those between the crystal structures (Figure 1A). Similar results are obtained when using the main chain atoms or all atoms for comparison (Appendix A). Thus, in terms of the RMSD, the AF structures are structurally similar to the crystal structures at all levels of structural details, including the C_α_ trace, the backbone, and all atoms, including the side chains.

Hierarchical clustering using the pairwise RMSD values as dissimilarity measures further indicates the AF structures structurally being similar to the crystal structures (Figure 1B and Appendix A). Our null hypothesis is that the AF structures are an outlier group and clustered into a distinct class being separate from the crystal structures. The purpose of the clustering is to reject this hypothesis, serving as proof of the accuracy of the AF structures by showing the differences between the AF and crystal structures being no larger than the positional variations caused by different crystallization conditions and different crystal symmetries. We found that the AF structure of neither SMYD2 nor SMYD3 is classified as an outlier group. In SMYD3 clustering, there are two major clusters at a height of 1 Å of RMSD, and three clusters at 0.8 Å of RMSD (Figure 1B). The clustering results are well correlated with the crystal lattices in which the proteins were crystallized. The currently known SMYD3 crystal structures were determined in four different crystal lattices, with two *P*2_1_2_1_2_1_ space groups of different unit cell parameters, one *P*2_1_, and one *P*6_1_ (Appendix A). Our analysis of the crystal packing indicates that these four crystal lattices represent four different types of crystal packing arrangements (Appendix A). Correlating with these arrangements, the two *P*2_1_2_1_2_1_ space groups are separated into two distinct clusters, while the third cluster is represented by the *P*2_1_ structures, which contain two molecules per asymmetric unit, in contrast to the other space groups with only one molecule per asymmetric unit. The *P*2_1_ is being clustered closer to the second *P*2_1_2_1_2_1_ group than the first one, while *P*6_1_ is clustered with the *P*2_1_ group. This result indicates the structural dissimilarities between the crystal structures being strongly associated with their crystal packing arrangements. The AF structure is clustered within the first *P*2_1_2_1_2_1_ group, the largest cluster that contains 72% of known SMYD3 structures and contains the highest-resolution structures determined to date. The higher the resolution of a structure, the higher the accuracy in defining the position of atoms, and, thus, the less errors there are in the structure. If the clustering could show an association with the resolution, one speculation is that the clustering might provide an estimation of how accurately the AF structure is predicted in terms of the resolution. To test this hypothesis, the association between the resolution and the deviation of the crystal structures from the “true” structure was estimated by using the highest-resolution structure, 6P7Z (1.19 Å), as reference, under an assumption of this structure representing the most accurate SMYD3 structure. We found that the resolutions are well correlated with the RMSD values, with a correlation coefficient (cc) of 0.66 (*p* = 0.00036) when using all crystal structures in the analysis (Figure 1C). However, if conditional on the crystal lattice, i.e., only using the structures within the first *P*2_1_2_1_2_1_ cluster for analysis, which eliminates the effect of the crystal lattice, there is a weak correlation, but it is not statistically significant (cc = 0.27, *p* = 0.28) (Figure 1C). This result further indicates the crystal lattices playing a major role in the clustering, while the resolution has little influence. Being clustered within the first *P*2_1_2_1_2_1_ group indicates the prediction errors of the AF structure being no more than the deviations found between the different crystal lattices. Similar results are observed in SMYD2 clustering, in which the crystal lattices are also strongly associated with the clustering (Figure 1B and Appendix A). Among twenty-two known structures, SMYD2 was crystalized in five different space groups, with a total of eight different types of crystal packing arrangements (Appendix A). About 78% of these structures have either *I*_4_ or *P*2_1_2_1_2_1_ space groups, and the rest of the structures have three structures in *C*_2_, one in *P*4_2_, and two in *P*2_1_. The AF structure is clustered with the *I*_4_ space group, which contains about 50% of known SMYD2 structures. This again indicates that the AF structure shows smaller deviations from the crystal structures than those caused by different crystal packing arrangements.

To further evaluate the reliability of the AF structures, we developed an inter-residue distance map (IRDM)-based metric, which, in contrast to the RMSD, is superimposition independent. In this method, the coordinates of the C_α_ atom of each residue in the canonical three-dimensional space are re-indexed with the inter-residue distance maps into a higher-order dimension, with the new dimension being equal to the total number of residues in a protein. In essence, each residue is specified by its distances to all residues, and each residue is involved in specifying all residues. To evaluate structural dissimilarity, each pair of the corresponding residues is compared in this high-dimensional space, and the overall difference between the two structures is defined by the root-mean-square difference of their Euclidean distances. This IRDM-based method has no need for superimposition since the inter-residue distance maps are translation and rotation invariant, representing the relative spatial arrangement of the residues within a protein. Similar results were obtained as those obtained with the RMSD-based analysis (Figure 1D,E). The differences between the AF and crystal structures, measured by this IRDM-based metric, are no more than those between the crystal structures (Figure 1D). Hierarchical clustering shows similar grouping results, with the AF structure being grouped within the main *P*2_1_2_1_2_1_ group in SMYD3 and within the *I*_4_ group in SMYD2, not being as an outlier group (Figure 1E). This IRDM-based metric further supports that the AF and crystal structures of SMYD proteins are structurally similar.

The residue-wise structural comparison reveals the most different regions between the AF and crystal structures. In both RMSD- and IRDM-based metrics, the largest differences are located at the loop regions and termini of the proteins, such as in SMYD3, being the loop between the second and third helices in the CTD domain, the loop between the first and second helices in the post-SET domain, and the loop connecting the MYND and core-SET domains (Figure 2A and Appendix A). Generally, the loops and termini of a protein can assume different conformations under different contexts, and the flexible nature of these regions is often an underlying cause of structural variability. In agreement with this notion, the residue-wise structural differences between the AF structure and one crystal structure show a high degree of correlation with those between the AF structure and another crystal structure (Figure 2B,C and Appendix A). Moreover, there is a significant correlation in the structural differences between the AF and crystal structures and the structural differences between the two crystal structures. The AF confidence scores, which indicate the reliability of AF prediction, also appear to correlate with the degrees of structural variability, with the low scores being associated with the regions with the large structural differences (Figure 2D). Since the low AF confidence scores suggest possible unstructured and flexible regions in isolation [18], we assessed if there is a correlation between the AF confidence scores and the structural flexibility. The refined B-factors, which indicate the level of atomic vibration, derived from the experimental structure determination, were used to represent structural flexibility. We found that the AF confidence scores are negatively correlated with the B-factors; that is, the lower the confidence scores, the higher the structural flexibility of the residues (Figure 2E). In agreement with this correlation, the AF expected position errors, which indicate the positional accuracy of AF prediction [18], also show a high degree of positive correlation with the B-factors (Figure 2F). This result indicates that the AF confidence scores and the AF expected position errors can be used to indicate structural flexibility. Together, the RMSD- and IRDM-based metrics suggest that the AF structures are reliable enough to be used for the structural and functional analysis of SMYD proteins.

### 2.2. “Crab”-Like SMYD5 Structure with a Large Deep Negatively Charged Cleft

SMYD5 is the most unique member of the SMYD protein family at both the sequence and structure levels. At the primary sequence level, the unique features include the lack of a conserved C-terminal domain (CTD), the large insertions in the SET and MYND domains, a poly-glutamic acid tract (poly-E) at the C-terminus, and an extra N-terminal sequence predicted to be a mitochondrial targeting sequence (Figure 3A). These unique sequence features are the fundamental determinants of the structural differences between SMYD5 and other SMYD proteins in terms of their overall shape and the structure at the active site (Figure 3B,C). The overall structure of SMYD5 resembles a “crab” with two large “legs” that are enriched with negatively charged residues (Figure 3B). The body of the crab is made up of four conserved domains, including the SET, MYND, post-SET, and SET-I domains, while the crab legs are formed by the structural features unique to SMYD5. For the legs, the thinner one is formed by the poly-E tract that forms a single helical structure, while the thicker one is formed by two large insertions from the MYND domain (M-insertion) and the SET domain (S-insertion). The M insertion is located in the middle of the MYND domain between the first and second pairs of zinc-chelating residues, about twenty-eight residues long. The S-insertion located within the SET domain between the β7–β8 hairpin, a hairpin involved in forming the substrate binding cleft in other SMYD proteins, consists of about thirty-nine residues. The most striking structural feature is that these two insertions protrude out from these domains and bundle together to form a new subdomain (SMI), which is not seen in any other SMYD proteins. Another profound feature is that on one side of this subdomain is a rather flat surface enriched with Glu/Asp residues (Figure 3D), and this flat surface, together with the poly-E tract, forms a large, deep negatively charged cleft, with the substrate binding site located at the bottom of this cleft. The cleft is about 35 Å deep, with a nearly uniform width of about 25 Å from the top to the bottom. Given its shape and charge, this cleft could provide a potential binding site for positively charged molecules. We previously showed that SMYD5 directly interacts with protamine, an arginine-rich protein that replaces histones during sperm chromatin condensation at the late stage of spermatogenesis [14]. SMYD5 was also shown to interact with and methylate histone H4, which is also a highly basic protein [11].

A poly-E sequence usually adopts a helical structure or is unstructured but can assume a fixed structure upon binding to other molecules. In some cases, the poly-E can mimic the structure of the DNA phosphate backbone involved in the regulation of gene transcription [19]. A structured poly-E region has been shown to be involved in intra- or intermolecular interaction with basic residues [19]. For instance, several histone chaperones, including proteins involved in sperm chromatin decondensation, consist of stretches of Glu/Asp residues that are known to be essential for histone binding [20]. In SMYD5, the poly-E is predicted as a helix in all species with available AF structures (Figure 4A). In these structures, the poly-E is rod-shaped, together with the acidic C-terminal end, and it forms a long negatively charged tail, extending away from the protein. However, all five available AF structures have different poly-E conformations, and this is in agreement with the fact that the poly-E in each of these structures is among the regions with the least confidence scores (Figure 4B and Appendix A). When these structures are superimposed, the orientations of the poly-E helices have a radial arrangement, appearing to swing with respect to one another in a plane roughly parallel to the negatively charged surface in the SMI subdomain. The extent of the swings varies between 5 and 180°, with Thr391 or Ser392 as hinge point. As a result, the conformations of the poly-E helices can be essentially clustered into two major orientations, with one pointing to the SET-I domain and the other to the opposite direction. This difference indicates that the poly-E tract in SMYD5 is flexible in nature and might not have a defined conformation without binding to other molecules.

In contrast to the poly-E tract, the structure of the SMI subdomain is relatively rigid, and, in all five AF structures of SMYD5, the locations and structures of this domain are highly superimposable (Figure 4C). The pairwise RMSD values calculated for this domain between these AF structures range from as low as 0.2 to 0.4 Å, while the locations of this domain relative to the SET domain remain largely unchanged, with a less than 3° difference between different structures. The rigidity of this domain is likely due to its double-layered structure that is formed by stacking interaction between the M-insertion and S-insertion (Figure 4D). There is a large amount of solvent-accessible surface area (565.6 Å^2^) buried at the interface between these two insertions, and the stability of their interaction is maintained by numerous hydrophobic contacts that hold them together. Because of the structural restraints imposed by this interaction, both M-insertion and S-insertion maintain a flat rigid structure. In addition, the residues at the interface of these two insertions are well conserved from the human to fly (Figure 4E), further supporting a stable, conserved interaction between them. We previously showed that deleting the M-insertion from SMYD5 caused the increased exposure of hydrophobic residues and dramatically reduced the thermal stability of SMYD5 [14]. Of particular interest is that the thermal stability of this mutant can be completely rescued by protamine binding, which can be performed without changing the level of hydrophobic exposure [14]. This result indicates that the protamine binding and the M-insertion might be functionally related in the context of stabilizing the SMYD5 structure.

### 2.3. The Flexible N-Terminal Nuclear Localization Signal

The N-terminal sequence is unique to SMYD5 and absent in other SMYD members (Figure 3A). This sequence, predicted to be a mitochondrial targeting sequence, is actually a novel non-classical nuclear localization signal (NLS) (Figure 5). Subcellular localization of SMYD5 was probed by using GFP as a reporter that was tagged to the C-terminus of the protein (Figure 5A). We found that intact SMYD5 is localized to the nucleus in all three cell lines examined, including HEK293, U2OS, and RAW264.7 (Figure 5B). When the first eighteen residues are removed, SMYD5 fails to concentrate to the nucleus; instead, there are more proteins present in the cytoplasm. To rule out possible GFP localization artifacts, the SMYD5 subcellar localization was also investigated by using Myc-tagged SMYD5 by immunostaining of the Myc tag, using an anti-Myc antibody. Similar results were obtained, with intact SMYD5 being localized in the nucleus and the N-terminus-deleted one being predominantly in the cytoplasm. In neither case was SMYD5 found to co-localize with the mitochondrial markers, including the MitoTracker and COX4, a mitochondria-localized protein. Different from the N-terminal COX4 sequence, which is able to drag GFP into the mitochondria, the N-terminal sequence of SMYD5 when attached to GFP is sufficient in itself to localize GFP into the nucleus (Figure 5B,C). This result suggests that the N-terminal sequence of SMYD5 contains a nuclear localization signal, which is not a mitochondrial targeting sequence. Classical NLS motifs are characterized by a stretch of basic residues arranged in monopartite or bipartite clusters [21]. The N-terminal sequence of SMYD5 is clearly not a classical NLS, because it does not contain any basic amino acids (Figure 3A). Overall, the first eighteen residues of SMYD5 are rather hydrophobic, with eleven of them, or about 61%, being hydrophobic and nonpolar residues, compared to an average of 45% of such residues in proteins. This sequence, together with the sequence between residues Ala19 and Val23, forms a helix–turn–helix structure, which packs against a small β-sheet, formed by strands β1 and β2, in the S-sequence (Figure 5D). This packing interaction leads to the formation of a hydrophobic core at the packing interface between the N-terminal sequence and the strand β1 that involves the residues Met5, Val8 and Phe9 from the N-terminal sequence and the residue Val25 from the strand β1. As a result of this interaction, the N-terminal sequence of SMYD5 is partially buried and not fully accessible in its current conformation. However, a comparison of the SMYD5 structures of different species, including human, mouse, rat, and zebrafish, indicates that the N-terminal sequence of SMYD5 is relatively flexible (Figure 5E). In all species, the N-terminal sequence is among a few regions with the low confidence scores produced by AF (Figure 4B). Although it occupies the same location and has a nearly identical amino acid sequence in these species (Figure 5F), the first seven residues of this sequence adopt four different conformations, with each being represented by a species. For instance, the conformation of the N-terminal sequence in the mouse is less structurally ordered, with a lower number of helical structures compared to that in the human structure. Because the AF confidence scores are correlated with the B-factors of the known crystal structures of SMYD proteins (Figure 2E), this indicates that the N-terminal sequence of SMYD5 is relatively flexible or structurally disordered, and this might facilitate its recognition for SMYD5′s cytonuclear transport.

### 2.4. Less Positively Charged MYND Domain

The MYND domain is a zinc finger motif that is organized around two zinc atoms chelated by seven cysteines and one histidine in a C4C2HC format (Figure 3A and Figure 6A). The core structure of the MYND domain is conserved in SMYD5, with a similar zinc chelating topology to that of other SMYD proteins. In SMYD1–3, the MYND domain is formed by an L-shaped kinked helix and an antiparallel β-hairpin nestled between the two arms of this kinked helix. In SMYD5, the kinked helix is broken down into two separate helices, αG and αH, due to one residue insertion (Tyr131) at the point of kinking (Figure 6A). While the helix αG aligns well with the first half of the kinked helix, the helix αH is much shorter in length than the second half of the kinked helix. Despite this difference, the residues that chelate the zinc atoms, Cys119 and Cys123 from helix αG and His132 and Cys136 from helix αH, are positioned very similarly to those from the kinked helix in other SMYD proteins. The largest difference in the MYND domains between SMYD5 and other SMYD proteins is the M-insertion, which is unique to SMYD5 (Figure 6A). This long insertion forms three helices, one long (αD) and two short ones (αE and αF), arranged in a rather flat triangular shape (Figure 6B). As it protrudes away from the core of the MYND domain, there is almost no intramolecular interaction between this insertion and other parts of the domain. As a result, the M-insertion in SMYD5 does not appear to contribute to or interfere with the structural and functional features that are conserved in the MYND domain.

The MYND domain in other SMYD proteins is known to interact with a proline-rich sequence with a PXLXP motif or may be involved in DNA binding [22,23,24]. For PXLXP interaction, the central Leu in the peptide was predicted to insert into a shallow surface pocket formed by Tyr73, Gln79, and Trp83 (SMYD1 numbering), with Tyr73 forming the base of the pocket [1]. These residues are completely conserved in SMYD2 and SMYD3, while, in SMYD5, they are partially conserved, with the residue Tyr118 conserved at the position of Tyr73, with Arg124 replacing Gln79, and with Thr128 replacing Trp83 (Figure 3A and Figure 6B). We previously showed that the SMYD3 interactome is significantly enriched with the PXLXP motif, which suggested that SMYD3 via the MYND domain interacts with some of its interacting partners [25]. Motif scanning by using ScanProsite [26] also reveals significant enrichment of the PXLXP motif in the SMYD5 interacting proteins (Appendix A). A total of thirteen matches were found in the SMYD5 interactome of nineteen proteins, i.e., the average matches being 0.684 per protein. As the chance for a random match for this motif is 0.103 per protein, this indicates that there is about a 6.6-fold enrichment of the PXLXP motif in the SMYD5 interacting proteins, and this is similar to that for the SMYD3 interacting proteins [25]. Thus, the enrichment of the PXLXP motif in the SMYD5 interacting proteins appears to not be affected by the partially conserved binding site in SMYD5 relative to that in SMYD3.

However, a major difference in the MYND domain between SMYD proteins is the number of positive charges on the domain (Figure 6D). The MYND domain has been shown to interact with DNA in SMYD3, as is in agreement with the fact that this domain in SMYD3 has a highly positively charged surface [1,24]. However, the MYND domain in SMYD5 is less positively charged compared to other SMYD proteins. The number of basic residues in the MYND domain are six, nine, eight, and fifteen for SMYD5, SMYD1, SMYD2, and SMYD3, respectively. We found that the DNA-binding ability of SMYD proteins appears to correlate with the number of basic residues in the MYND domain (Figure 6E). SMYD3 causes the largest level of DNA shift in an electrophoretic mobility shift assay, followed by SMYD1 and SMYD2, while, under the same experimental condition, SMYD5 shows no apparent DNA binding. The difference in the number of positively charged residues suggests that the MYND domain of SMYD5 may have evolved into a different branch separate from other SMYD family members. Bayesian inference of the phylogeny of the MYND domains indicates that SMYD1–4 are grouped together, whereas SMYD5 is classified as a separate clade with a high probability (Figure 6C and Appendix A). A similar tree topology was obtained by using the maximum-likelihood-based phylogenetic analysis (Appendix A). This result indicates that, in terms of the MYND domain, SMYD5 is more divergent from other SMYD members than those members are from each other.

### 2.5. Evolutionarily Conserved SET Domain Largely Structurally Conserved

In SMYD proteins, only the MYND domain is not involved in forming the active site, while other domains, including SET, SET-I, post-SET, and CTD, are either involved in cofactor binding, substrate binding, or both. Among these domains, the SET domain is the most structurally conserved. This domain is split into the S-sequence and the core-SET domain by the MYND and SET-I domains at the primary sequence level, while, in structure, the split S-sequence and core-SET domain come back together to form an intact evolutionally conserved SET fold (Figure 7). The S-sequence forms four β-strands (β1–β4) and provides an Arg/Lys residue for stacking against the adenosine moiety of a bound cofactor. In SMYD5, the S-sequence is highly conserved in sequence and structure, with only one residue insertion (Ser30) within the loop between the first and second β-strands. For the core-SET domain, all secondary structures and core structural motifs within this domain are well aligned between SMYD5 and other SMYD proteins. These include the NH motif at the base of the cofactor binding site and a cysteine residue (Cys318) that mediates the interaction between the SET and post-SET domains via chelating a zinc atom. The largest difference is the S-insertion, which is unique to SMYD5, but this insertion protrudes away from the SET domain and stays distal to the active site.

The post-SET domain intimately interacts with the SET domain due to being bundled together by a zinc center that involves chelating residues from both of these domains (Figure 7). The post-SET domain, generally consisting of one long and two short helices, is involved in the formation of the cofactor binding site, the substrate binding site, and the target lysine access channel. In the post-SET domain, a completely conserved Phe, which stacks against the adenosine moiety of the cofactor, is located within a loop connecting the first long helix and the first short helix. A Tyr residue, another essential aromatic residue, lines the target lysine access channel, located near the end of the first long helix. The loop preceding the post-SET domain is involved in the formation of the binding cleft for a substrate peptide. The post-SET domain of SMYD5 is structurally similar to that of other SMYD proteins, including the two essential aromatic residues (Tyr372 and Phe374) and the zinc-chelating residues (Cys376, Cys378, and Cys381) that hold the SET and post-SET domains in contact. One difference is an additional helix (αS) before the first long helix due to a three-residue insertion in SMYD5. The second difference is that SMYD5 does not form the third helix that is located near the junction between the N- and C-lobes in other SMYD proteins. As a result, after the second helix, the post-SET domains in SMYD proteins diverge in different directions: in SMYD5, it goes straight to the flexible poly-E tract, whereas, in other SMYD proteins, it folds backward and then goes into the CTD domain.

### 2.6. Larger SET-I Domain Makes the Active Site Even More Buried

The SET-I domain of SMYD5 is larger than that of other SMYD proteins in size, containing several insertions (Appendix A). This domain is about 110 amino acids long in SMYD5, compared to about eighty-five residues in SMYD1–3. The SET-I domain is an all-helix structure, generally consisting of three long and three short helices (Figure 8A). All short helices are located between the first and second long helices, where the region between the second and third short helices contributes to the formation of the cofactor binding site. In SMYD5, there is a seventeen-residue-long insertion at the N-terminus of the SET-I domain. This long insertion adds an extra helix (αI) to the domain, thus leading the polypeptide chain to run over the β7–β8 hairpin. Such a structure, not seen in other SMYD proteins lacking this insertion, leads to a deeper substrate binding cleft in SMYD5 (Figure 8B). Based on the superimposed ERα peptide from the SMYD2–ERα complex structure, the loop connecting this new helix and the original first long helix could be involved in interaction with the substrate (Figure 8A,B).

Another difference is seen at the first and second short helices, which, in SMYD5, are combined into a single long helix (αK) (Figure 8A). This combined helix follows the helical direction of the second short helix in other SMYD proteins, with the C-terminal half of this combined helix aligned with the second short helix, but with its N-terminal half showing a large shift from the first short helix by as much as 8 Å (Figure 8C). This large shift appears to be related to the NLS sequence at the N-terminus of SMYD5, as this sequence occupies the position equivalent to a position occupied by the first short helix in other SMYD proteins. Another large difference is found at the third short helix, where, in SMYD5, there is an insertion of four glutamic acid residues (E-insertion) (Figure 8A). AF predicted that this insertion can adopt two major conformations, with one conformation being a helical structure and the other conformation being a loop structure (Figure 8A and Appendix A). This prediction is consistent with the fact that the E-insertion might be structurally flexible with the relatively low AF scores (Figure 4B). With the helical conformation, the E-insertion extends the length of the third short helix, yielding a longer helix that extends from the cofactor binding site to the substrate binding site (Appendix A). With the loop conformation, the E-insertion displaces the third short helix toward the substrate binding site, while the protruding-out conformation of the E-insertion causes the cofactor binding site to be more buried in SMYD5 than in other SMYD proteins (Figure 8D). We previously showed that SMYD1–3 have a nearly buried cofactor binding site compared with that of other SET domain-containing proteins, such as SET7 and Dim-5 [1,15,16,17]. In SMYD5, the loop conformation of the E-insertion makes the cofactor binding site even more buried. However, the flexible nature of the E-insertion suggests that it might be able to undergo a conformational change during cofactor turnover. Despite those differences, the second and third long helices, which are involved in forming the active site, align well between SMYD5 and other SMYD proteins, with no gaps or insertions in the alignment. The loop between the second and third short helices that is involved in forming the cofactor binding site is also well aligned between SMYD proteins.

### 2.7. Incomplete Aromatic Channel in the Active Site

The active site of SMYD proteins can be divided into three structural units, namely the cofactor binding site, the substrate binding site, and the target lysine access channel (Figure 9). The methyltransferase reaction occurs at the junction between the cofactor binding site and the target lysine access channel, with the latter connecting the former to the substrate binding site. There are four regions from four domains involved in forming the cofactor binding site (Figure 9A). The walls of the cofactor binding site are formed by the loop between the first and second β-strands in the S-sequence, the loop between the second and third short helices in the SET-I domain, and the loop connecting the second and third helices in the post-SET domain, while the base of this site is formed by a loop following the only helix in the core-SET domain. Despite a few differences in sequence, all of these regions are structurally well aligned between SMYD5 and other SMYD proteins. However, with respect to the substrate binding site, there are several notable differences between SMYD5 and other SMYD proteins (Figure 9B). In SMYD proteins, the substrate binding-site cleft is formed by the β8–β9 hairpin and a loop connecting the SET and post-SET domains. However, without the CTD domain, the substrate binding site in SMYD5 is widely open at one end, in contrast to that in SMYD1–3 being closed at both ends by the SET-I and CTD domains. In addition, this binding cleft in SMYD5 is slightly wider in size than that in other SMYD proteins, due to the substitution of two substrate-facing residues in the β8 strand by the small glycine residues. Another difference is the location of a substrate-binding Glu residue. In SMYD2, Glu187 in the β8 strand is involved in substrate binding via its side-chain carboxylate group, while, in SMYD3, Glu192, situated in the β9 strand, performs a similar role [9,27]. SMYD5 does not have an equivalent Glu residue at either of those locations; instead, the side chain of Glu303, located at a position next to Glu192 of SMYD3 in the β9 strand, points to the direction where the substrate binds. Thus, these Glu residues, though having different locations in the β8–β9 hairpin, may have a complementary role in substrate binding.

One of the most surprising features in SMYD5 is the lack of a tri-aromatic residue arrangement in the target lysine access channel (Figure 9C). A target lysine is usually held in position in the active site by three aromatic residues, which form a narrow channel with a dimension matching the size of the lysine side chain [1,9]. In SMYD1–3, the first aromatic residue comes from the loop connecting the SET and post-SET domains, the second one is provided by the post-SET domain from its first long helix, and the third one is from the region at the beginning of the β8 strand. In SMYD5, the first two such residues (Tyr351 and Tyr372) are conserved, but not the third one, which is replaced by the non-aromatic residue Gln254. Our survey of all SET domain-containing proteins that have been characterized to have a lysine methyltransferase activity contains an aromatic residue at this position. Mutation of the equivalent residue Phe184 in SMYD2 completely disrupted the methyltransferase activity [28]. SMYD5 was previously identified as a histone H4K20 methyltransferase, and this methylation was found to be associated with the decreased expression of several TLR4 target genes, likely regulating the higher-order structure of chromatin [11]. However, one recent study showed that SMYD5 does not methylate histone H4; instead, it catalyzes the methylation of histone H3 at K36 and K37 [29]. It was shown that knockout of SMYD5 in the murine embryonic stem cells partially reduced the global methylation level of H3K37, suggesting that SMYD5 is one of histone lysine methyltransferases catalyzing H3K37 methylation in vivo. We found that, compared with SMYD2 or SMYD3, SMYD5 only shows a background level of methylation on histones H3 and H4 in our assay conditions, but SMYD2 efficiently methylates both of these proteins (Figure 9D).

## 3. Discussion

Our rigorous structural validation of AF structures sets an example of how to evaluate AF structures and to determine what level of structural details can be trusted for structural analysis. In the case of the SMYD protein family, the AF structures can be reliably used at all levels of details, from the C_α_ trace, the backbone, to all atoms, including the side chains. The reliability of AF structures was validated by a classic RMSD-based metric and a new metric based on inter-residue distance maps. The latter metric developed in this study provides an additional tool to help analyze dissimilarities between the AF and crystal structures of proteins. As a superimposition independent method, the IRDM-based metric allows for a topological representation of a protein independent of its absolute position and orientation, therefore reflecting the true structural differences in structural comparison, which could otherwise be compensated by the process of superimposition [30]. The premise of this method is in agreement with the coherent nature of a protein structure being as an interconnecting network system involving both short- and long-range communications [31]. The reliability of using the AF structures in analyzing the SMYD5 structure was also supported by the correlation between the structure and biochemical evidence, which consequently provides the structural basis for the functional divergence of SMYD5. The AF confidence scores as a valuable indicator of structural flexibility also help provide further insights into the relationships between SMYD5 structure and function.

SMYD5 structure resembles a “crab” with the two large “legs” enriched with negatively charged residues (Figure 3). The fundamental determinants of this crab-like structure are the C-terminal poly-E tract and the new subdomain SMI formed by the large insertions in the SET and MYND domains. These two regions, unique to SMYD5, define the structure of SMYD5 as its own kind, being distinct from the classical two-lobed structure seen for other SMYD proteins. One striking feature here is the large deep cleft formed between these regions that has the negatively charged surfaces on both sides. The size, shape, and charge of this cleft suggest a potential binding site for basic molecules, such as protamines and histones, which are known to interact with SMYD5. On the other hand, this large, deep cleft helps explain how SMYD5 manages to form the substrate binding site without the conserved CTD domain. The CTD domain in other SMYD proteins is involved in binding to a substrate by sandwiching the substrate between the two lobes [9]. The structural comparison indicates that the missing CTD domain is actually partially compensated by the new SMI subdomain, which occupies a position that overlaps with the top part of the CTD domain, thereby contributing to the formation of the substrate binding site (Figure 3C). Among all unique structural features, the most surprising one is the incomplete aromatic channel used for the target lysine binding. The target lysine access channel usually has a width perfectly matching the thickness of the lysine side chain, so that the lysine can be held stably in place, facilitating the methyl transfer from the methyl donor to its epsilon nitrogen [1]. The solution to such a perfect fit is provided by the evolutionally conserved tri-aromatic residues, which are arranged as a triangular prism, with each of its lateral faces made of one aromatic ring, thus creating a narrow channel running through the middle of the prism. The absence of an aromatic residue at the position equivalent to Gln254 makes the target lysine access channel in SMYD5 much wider and less confined (Figure 9C). The wider channel raises a question of how SMYD5 maintains a lysine side chain in the channel in a stable bound conformation and avoids its wobbling during the reaction. One possibility is a nearby aromatic residue that could flip into the position to make a complete channel. There are, however, no suitable aromatic residues that can assume such a role by simply adopting a different rotameric state, and the nearest aromatic residue (Phe308), which is conserved in SMYD proteins and is involved in forming the core of the SET domain, is 7.5 Å away from the channel, a distance that is a bit far to reach (Figure 9C). Another possibility is that the target lysine access channel could be completed by an aromatic residue from the substrate; that is, the channel can only be completely formed upon substrate binding with a donation of an aromatic residue from the substrate. Such an idea is not imaginary, as in the histone H4K20 lysine methyltransferase SET8, a histidine residue, being two residues upstream the target lysine, in the substrate H4 involved itself in forming the target lysine access channel [32].

The extra N-terminal sequence is another unique feature of SMYD5. If only based on the primary amino acid sequence, this sequence was predicted as a mitochondrial targeting signal, rather a nuclear localization signal (Appendix A). We demonstrated that this N-terminal sequence encodes a novel non-classical NLS, which does not resemble any classes of known NLS motifs. A classic NLS normally contains a single stretch of basic amino acids or two such separate sequences arranged in a bipartite manner [21]. The N-terminal sequence of SMYD5, which is neither similar to these canonical motifs nor rich in arginine or lysine residues, thus defines a novel non-classical NLS of its own category. Because this sequence is sufficient by itself in localizing GFP into the nucleus, it is unlikely that it is a part of a bipartite NLS. In the bipartite NLS, the upstream and downstream clusters of amino acids are interdependent and indispensable, which jointly determine the localization of the protein in the cell [21]. In conclusion, the unique SMYD5 features, including the lack of the CTD domain, the N-terminal non-classical NLS sequence, the large negatively charged cleft, the new SMI domain, the less positively charged MYND domain, and the C-terminal poly-E tract, have expanded our understanding of the SMYD protein family in their structure-and-function relationships. The approach developed for the rigorous structural validation of AF structures also sets the stage for the broad use of AF structures for a reliable structure-and-function analysis.

## 4. Methods

### 4.1. Molecular Cloning

For methyltransferase assays and DNA gel shift experiments, full-length human SMYD constructs, including SMYD1, SMYD2, SMYD3, and SMYD5, were built by subcloning their DNA fragments into the pCDF-SUMO vector containing an N-terminal 6xHis-SUMO tag [15]. For immunofluorescence experiments, C-terminal Myc-tagged full-length human SMYD5 (hSMYD5(FL)-Myc), full-length mouse SMYD5 (mSMYD5(FL)-Myc), and human SMYD5 lacking the first eighteen residues (hSMYD5(19-418)-Myc) were built by subcloning their DNA fragments into the pcDNA3.1 vector. For Green Fluorescence Protein (GFP) fluorescence experiments, C-terminal GFP-tagged constructs, including full-length human SMYD5 (hSMYD5(FL)-GFP), full-length mouse SMYD5 (mSMYD5(FL)-GFP), the first thirty residues of human SMYD5 (hSMYD5(1-30)-GFP), human SMYD5 lacking the first eighteen residues (hSMYD5(19-418)-GFP), the mitochondrial targeting signal of human COX4 (hCOX4(1-25)-GFP), were constructed by using the pEGFP-N1 vector. The corresponding primers used in molecular cloning are listed in Appendix A.

### 4.2. Protein Purification

Proteins with the 6xHis-SUMO tag, including SMYD1, SMYD2, SMYD3, and SMYD5, were expressed and purified according to the procedures described previously [9]. The 6xHis-SUMO tag was cleaved off with yeast SUMO Protease 1, generating a native N-terminus. Histones H3 and H4 were purified from inclusion bodies under denaturing conditions, according to the previously described procedure [33]. Briefly, BL21 cells that expressed histones were lysed by French Press. Inclusion bodies were isolated by centrifugation and resuspended in the denaturing buffer (7 M urea, 20 mM NaCl). Histones were purified from the inclusion bodies by the cation-exchange chromatography and then refolded by dialysis against the refolding buffer (10 mM NaCl, 1 mM β-mercaptoethanol).

### 4.3. Cell Cultures and Transfection

HEK293, U2OS, and RAW264.7 cells were obtained from the American Type Culture Collection (Manassas, VA) and maintained as described previously [34]. Briefly, the cells were cultured in the Dulbecco’s modified Eagle’s medium (DMEM) (Invitrogen, Waltham, USA) supplemented with 10% fetal bovine serum (FBS), 100 units/mL penicillin, and 100 g/mL streptomycin. The cells were maintained at 37 °C in 5% CO_2_ and routinely passaged at a ratio of 1:4 when 70–80% confluent. For immunostaining and GFP fluorescence experiments, cells were seeded in 6-well plates, with a cover slide inside each well. When reaching 80% confluency, cells were provided with fresh media, and transfection was carried out with the Lipofectamine 2000 (Invitrogen) transfection kit, according to the manufacturer’s instructions. After 48 h post-transfection, cell slides were rinsed with phosphate-buffered saline (PBS, Invitrogen) and then used for direct- or immuno-fluorescence analysis.

### 4.4. Immunostaining and GFP Fluorescence

For immunofluorescence, unstained slides were fixed by 4% paraformaldehyde for 15 min, followed by permeabilization with 0.2% triton X-100 for 15 min. The slides were then blocked with 5% bovine serum albumin (BSA) for 1.5 h at room temperature before applying primary antibodies against the Myc tag (1:200, Ab18185, Abcam, Cambridge, UK) and COX4 (1:200, Ab33985, Abcam). After PBS rinsing, secondary antibodies, including a donkey anti-rabbit antibody conjugated with Alexa Fluor 488 and a donkey anti-mouse secondary antibody conjugated with Alexa Fluor 594 (1:2000, Invitrogen, Waltham, MA, USA), were applied for 1 h at room temperature. For GFP fluorescence, unstained slides were incubated with the mitochondrial marker MitoTracker (Thermo Scientific, Waltham, MA, USA) for 50 min prior to fixation. In both cases, signal detection was performed after applying the mounting buffer ProLong Diamond Antifade Mountant with DAPI (Thermo Scientific). Representative images were taken with a Nikon Eclipse E800 with a CFI60optical system for adjusting brightness and contrast and image cropping.

### 4.5. Methyltransferase Assays

Antibody-based methyltransferase assay was carried out similarly to that described previously [15]. Reactions were performed by incubating SMYD proteins with histone H4 or H3 in the presence of AdoMet at 30 °C overnight in a reaction buffer containing 50 mM Tris pH 8.5, 25 mM NaCl, 5% glycerol, and 2 mM DTT. Lysine methylation was detected by Western blot analysis, using antibodies against mono-, di-, and trimethylated lysine (ab23366 and ab76118, Abcam). Enzymatic activities were quantified based on chemiluminescence, using a CCD gel imager. MTase-Glo methyltransferase assay was carried out according to the manufacturer’s instructions (Promega, Madison, WI, USA). Reactions were set up by incubating 1 μg SMYD proteins with 10 μg H3 or H4 in the presence of 500 μM of AdoMet overnight. After incubating the reactions with the MTase-Glo reagent and MTase-Glo Detection Solution, the amount of produced AdoHcy was determined by measuring luminescent signals, using a microplate reader. The reactions were quantified by using an AdoHcy standard curve that correlates the amount of luminescence to the AdoHcy concentration.

### 4.6. Electrophoretic Mobility Shift Assay

SMYD–DNA interaction was examined by using an electrophoretic mobility shift assay (EMSA). SMYD proteins were incubated with double-stranded DNAs at room temperature for 30 min before being subjected to agarose gel electrophoresis. Results were analyzed by ethidium bromide staining of DNA. DNA probes used in the experiments include a 44-bp oligonucleotide (5′-TTACGCCCTCCTGAAACTTGTCATCCTGAA-TCTTAGAGGGGCCC-3′) and a 24-bp oligonucleotide (5′-CACCGCTCACGAAACGGACTTCCA-3′, 5′-AAACTGGAAGTCCGTTTCGTGAGC-3′).

### 4.7. SMYD AF and Crystal Structures

SMYD crystal structures were downloaded from the Protein Data Bank (PDB) (Appendix A). SMYD AF structures and their expected position errors were downloaded from the AlphaFold Protein Structure Database. AF prediction of SMYD5 structures was performed by using the AlphaFold CoLab notebook (Phenix version) [35].

### 4.8. RMSD and Inter-Residue Distance Maps (IRDM) Based Metrics

RMSD was calculated by using *lsqkab* from CCP4 [36]. IRDM was calculated by using R. IRDM is a **n** × **n** matrix, where **n** is the number of residues within a structure, and its entry values are the residue-wise Euclidean distances that are calculated by using the coordinates of C_α_ atoms. IRDM metric was calculated as the root mean of the sum of the squared differences between each corresponding pair of rows of two IRDM matrices. IRDM distances used in Figure 2B are the absolute values of the element-wise differences between two IRDM matrices.

### 4.9. Phylogenetic Analysis of MYND Domains

Phylogenetic analysis was performed on the MYND domains of human SMYD proteins. The MYND domain of ETO, a distantly related protein, served as an outgroup to root trees. Sequence alignment used for tree building was obtained by using ClustalX [37], and after being manually modified under the guidance of structure superimposition, gaps and insertions were excluded from the analysis. The trees were built by using Bayesian inference and the maximum likelihood algorithm. Bayesian inference was performed by using MrBayes [38]. There were four model parameters being estimated by Bayesian inference, namely the tree topology with a uniform prior, the branch lengths with an unconstrained compound Dirichlet prior, the shape parameter of the gamma distribution of among-site rate variation, and a mixture of fixed-rate amino acid substitution models with equal prior probability. Posterior probability distribution of these parameters was sampled by 1,000,000 cycles of Markov Chain Monte Carlo (MCMC) simulation with two independent runs. Burn-in periods of the simulation runs were determined based on when the log probability of observing the data started to plateau. Convergence of the model parameters was monitored by two diagnostics: ESS, Estimated Sample Size; and PSRF, Potential Scale Reduction Factor. Maximum likelihood (ML)-based phylogenetic analysis was performed by using *phangorn* in R [39]. Based on a maximum-likelihood-ratio test run over seventeen amino acid substitution models, *Dayhoff*, with the lowest AIC (Akaike Information Criterion) score, was chosen as the model of protein evolution. This result is in agreement with Bayesian inference, in which the posterior probability distribution of the amino acid substitution models was dominated by two models, with *Dayhoff* contributing about 84% and *vt* about 16%. In ML tree building, the topology was searched by using a stochastic nearest-neighbor interchange algorithm, and the edge lengths and the shape parameters of the gamma site rate distribution were optimized. The reliability of the trees was assessed by bootstrapping, using 1000 replications.

### 4.10. SMYD5 Interactome

SMYD5 interactome was assembled through manually curating SMYD5 interacting proteins retrieved from four protein–protein interaction databases, including GPS-Prot [40], BioGRID [41], HitPredict [42], and STRING [43]. GPS-Prot archives seven experimentally determined interactors from human, BioGRID sixteen, HitPredict eight, and STRING ten, making a total of twenty-four unique interactions. The final list of proteins used in this study was generated by removing the interactors with low-confidence reliability scores. Proteins were considered as low-confidence interactors if their interaction score is less than 0.3 in STRING. The final list contains nineteen interactors.

### 4.11. PXLXP Motif Enrichment Analysis

The presence of the PXLXP motif in SMYD5 interacting proteins was scanned by using ScanProsite [26]. Enrichment analysis of this motif was performed by comparing the number of matches to the expected random matches of the motif in the background, i.e., ~100,000 sequences or 50,000,000 residues [44]. Fold enrichment was calculated as the ratio of the average matches within our protein list to the average random matches against the background. For the PXLXP motif, the number of expected random matches is 10,318 in the background [44]. Significance of enrichment was assessed by using the Benjamini-and-Hochberg procedure, with a False Discovery Rate (FDR) < 0.05 being considered as significant.

### 4.12. Statistical Analysis

All statistical analyses, unless otherwise stated, were performed in R. Methyltransferase assay results were presented in the form of mean ± standard error (SE). The statistical difference between the means of activities was analyzed by one-way ANOVA if more than two groups were compared, and the post hoc comparisons on each pair of means were then assessed by using the Tukey Honest Significant Difference (HSD) test. A two-tailed Student’s *t*-test was used if only two groups were compared. The statistical difference between the means of RMSD values or IRDM values was tested by Welch’s *t*-test. The association between RMSD values and crystal resolutions, between two sets of RMSD values or IRDM values, or between the AF expected position errors and B-factors was measured with the Pearson correlation coefficient. The significance of the correlation coefficient was evaluated by t-statistic. The strength of association between the AF confidence scores and B-factors was analyzed by the Spearman rank correlation coefficient. In all cases, a *p*-value ≤ 0.05 was used as the criterion for statistical significance.

## Figures and Tables

**Figure 1 biomolecules-12-00783-f001:**
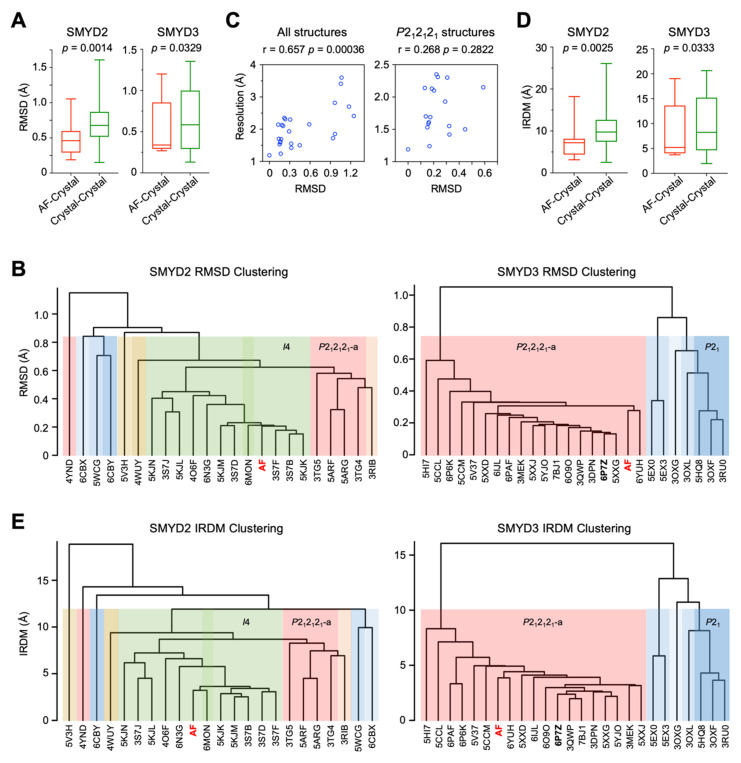
**Comparison of AlphaFold and crystal structures of SMYD proteins based on the C****_α_ trace.** (**A**) Boxplot of RMSD values between AF and crystal structures and between the crystal structures. (**B**) Hierarchical clustering using pairwise RMSD values as dissimilarity measures. Clusters are shaded in different colors according to space groups. (**C**) Scatter plot between the resolution of SMYD3 crystal structures and their RMSD to the reference structure 6P7Z. The analysis was performed with all SMYD3 crystal structures (left) and with the structures within the *P*2_1_2_1_2_1_-a cluster (right). (**D**) Boxplot of IRDM values between AF and crystal structures and between the crystal structures. (**E**) Hierarchical clustering using pairwise IRDM values as dissimilarity measures.

**Figure 2 biomolecules-12-00783-f002:**
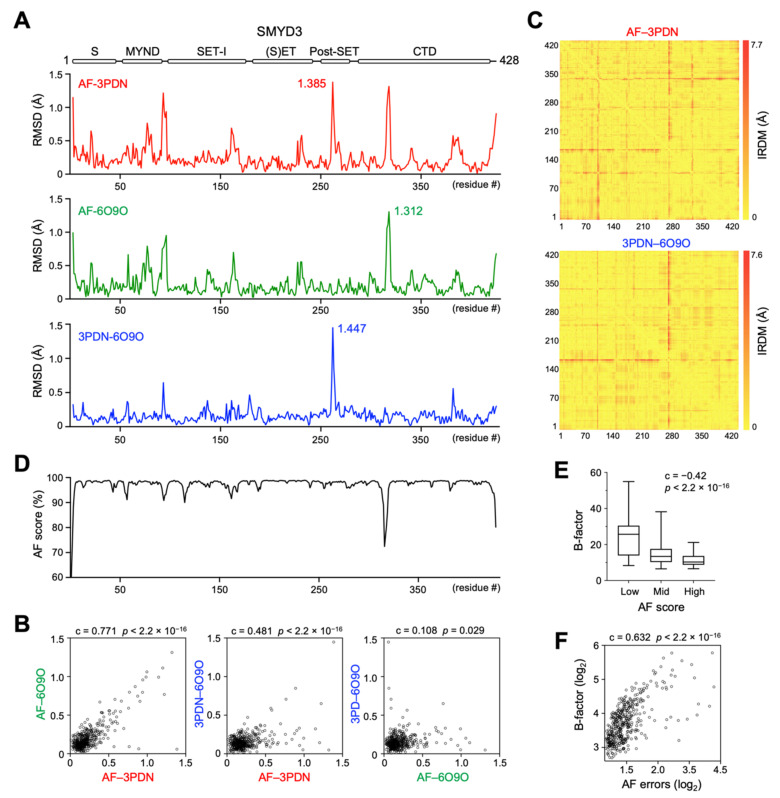
**Residue-wise structural comparison of SMYD3 AlphaFold and crystal structures.** (**A**) A plot of RMSD as a function of residue number between AF and 3PDN (top), between AF and 6O9O (middle), and between 3PDN and 6O9O (bottom). (**B**) Scatter plot of RMSD values between AF–3PDN and AF–6O9O (left), between AF–3PDN and 3PDN–6O9O (middle), and between AF–6O9O and 3PDN–6O9O (right). (**C**) Heatmap of IRDM distances between AF and 3PDN (top) and between 3PDN and 6O9O (bottom). (**D**) A plot of the AF confidence score versus residue number. (**E**) The AF confidence scores are negatively correlated with the B-factors of 3PDN. AF scores are divided into three categories: high (≥98.5), medium (between 98.5 and 96.0), and low (<96.0). (**F**) Scatter plot between the AF expected position errors and the B-factors of 3PDN.

**Figure 3 biomolecules-12-00783-f003:**
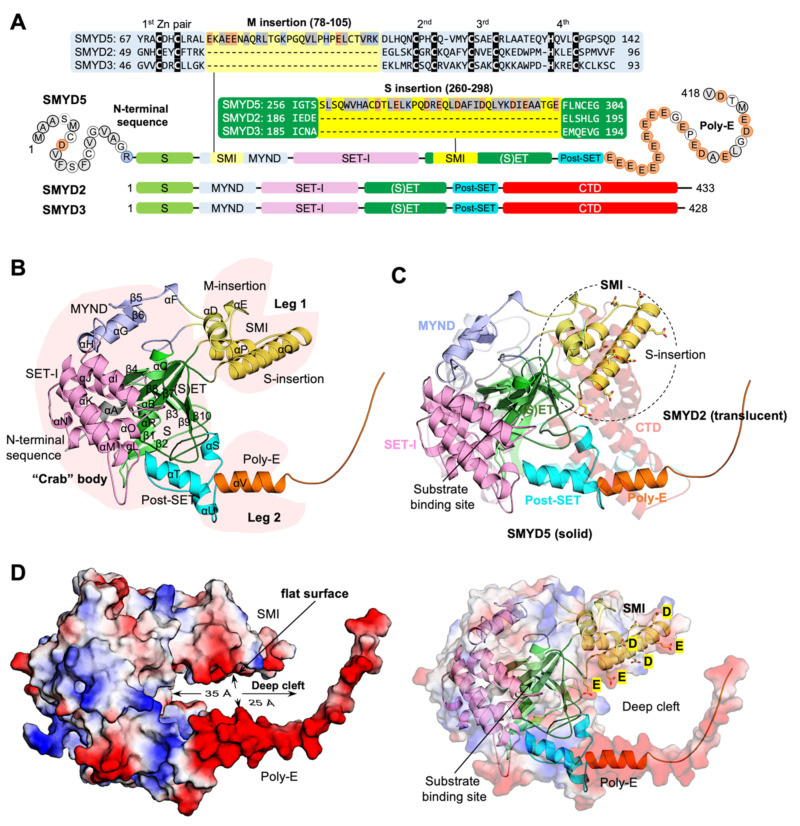
**“Crab”-like SMYD5 structure** (**A**) Domain structures and sequence alignment of SMYD proteins (SMYD5, SMYD2, and SMYD3). Acidic, basic, and hydrophobic residues are shaded in orange, blue, and grey, respectively. Zinc-chelating residues are shown as white on black. (**B**) Ribbon representation of SMYD5 overall structure. The N-terminal sequence, S-sequence, MYND, M-insertion, SET-I, core SET, S-insertion, post-SET, and poly-E are depicted in grey, light green, blue, yellow, pink, green, gold, cyan, and orange, respectively. Secondary structures, α-helices and β-strands, are labeled and numbered according to their position in the sequence. (**C**) Structural comparison of SMYD5 (solid color) and SMYD2 (translucent). (**D**) Surface representation of SMYD5 structure. The surface is colored according to the electrostatic potential: red, white, and blue correspond to negative, neutral, and positive potential, respectively.

**Figure 4 biomolecules-12-00783-f004:**
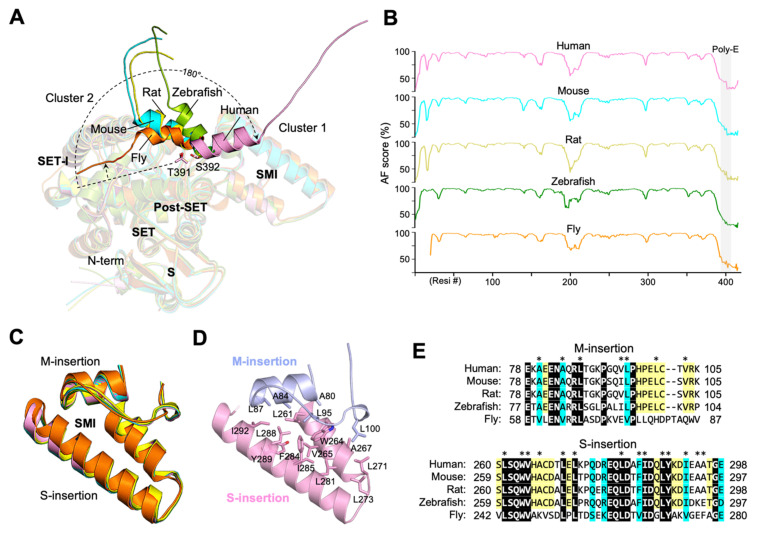
**Flexible Poly-E tract and rigid SMI subdomain.** (**A**) Superposition of SMYD5 AF structures at the poly-E tract across species. (**B**) A plot of the AF confidence score against residue number. (**C**) Structural comparison of SMI subdomains across species. (**D**) Hydrophobic interactions stabilize the double-layered structure of the SMI subdomain. (**E**) Sequence alignment of SMI subdomains across species. Hydrophobic residues at the interface of the M-insertion and the S-insertion are indicated by asterisks (*) above the alignment. Identical residues are shown as white on black, and similar residues appear shaded in cyan. The coloring scheme in A–E is according to species, with human, pink; mouse, cyan; rat, yellow; zebrafish, green; fly, orange.

**Figure 5 biomolecules-12-00783-f005:**
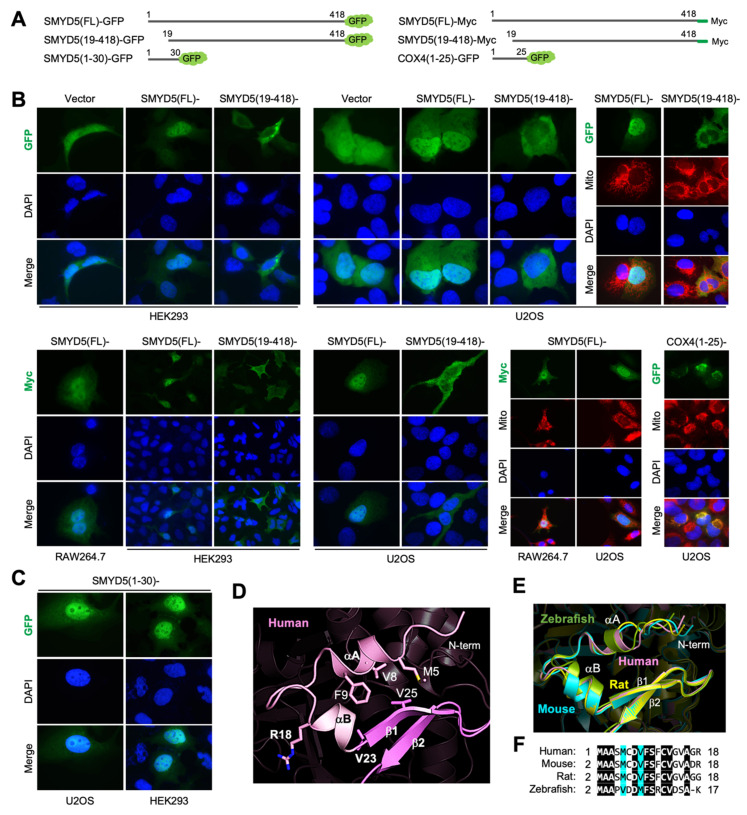
**Novel non-classical nuclear localization signal.** (**A**) SMYD5 constructs used in immunostaining and fluorescence microcopy. (**B**) SMYD5 subcellular localization. Similar results were obtained in U2OS, HEK293, and RAW264.7 cells. (**C**) The N-terminal sequence of SMYD5 is sufficient in itself to localize GFP into the nucleus. (**D**) Structure of the NLS of human SMYD5. (**E**) Structural comparison of NLS sequences across species, with human in pink, mouse in cyan, rat in yellow, and zebrafish in green. (**F**) Sequence alignment of NLS sequences across species, with completely identical residues shown as white on black, and similar residues appearing shaded in cyan.

**Figure 6 biomolecules-12-00783-f006:**
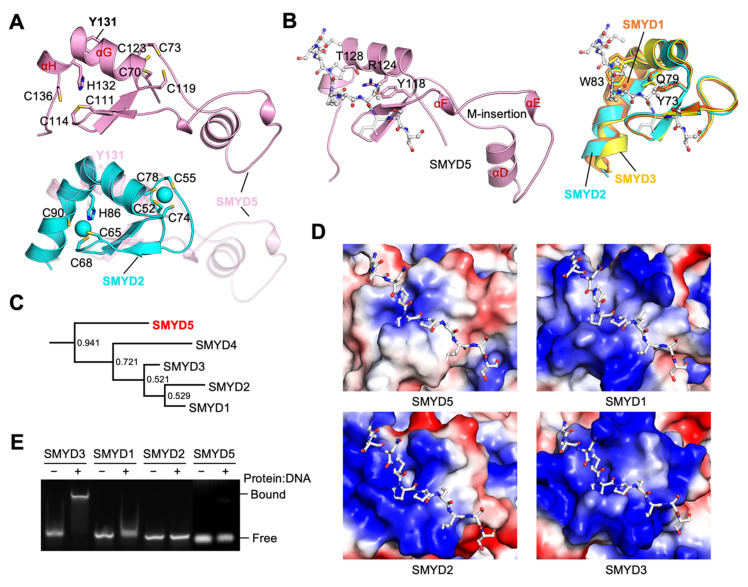
**Less positively charged MYND domain.** (**A**) Structure of MYND domains. SMYD5 (top) and its superposition with SMYD2 (bottom). (**B**) PXLXP motif binding pockets. SMYD5 (left) and superposition of SMYD1 (orange), SMYD2 (cyan), and SMYD3 (yellow) (right). A proline-rich peptide modeled based on the superposition with the structure of AML1/ETO (PDB code: 2ODD) is depicted by sticks. (**C**) Phylogenetic analysis of the MYND domains of human SMYD proteins using Bayesian inference. Node labels indicate the clade credibility or the posterior probability of branches. Branch lengths are scaled to the number of substitutions per site. (**D**) Surface representation of MYND domains of SMYD proteins, with the coloring scheme same as in Fig. 3D. (**E**) EMSA analysis of DNA binding of SMYD proteins.

**Figure 7 biomolecules-12-00783-f007:**
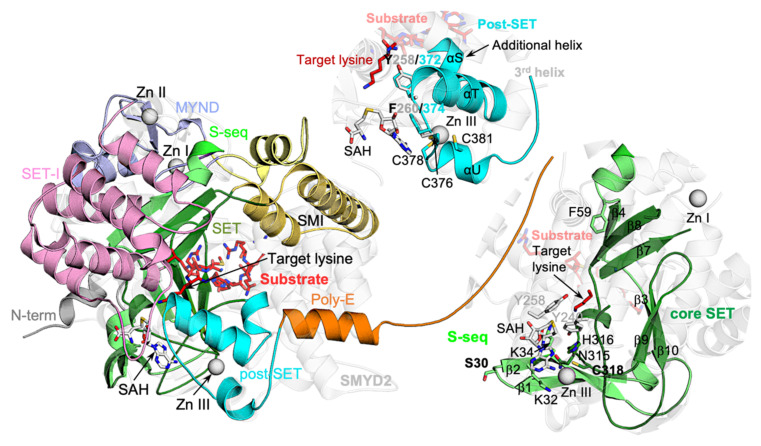
**Structural features evolutionally conserved in SMYD proteins.** SMYD5 structure with the cofactor SAH and substrate peptide modeled based on the superposition with SMYD2–ERα structure.

**Figure 8 biomolecules-12-00783-f008:**
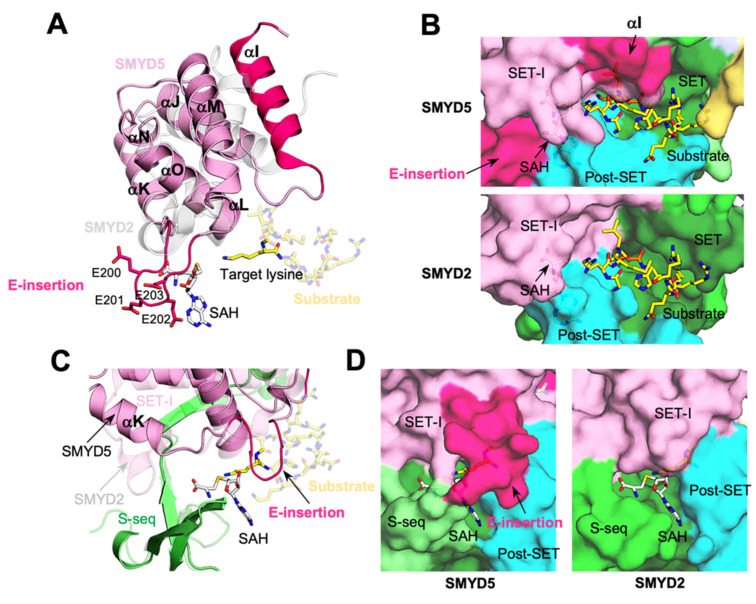
**Larger SMYD5 SET-I domain makes the active site even more buried.** (**A**) Structure of SET-I domains. SMYD5, pink; SMYD2, grey. (**B**) A deeper substrate binding site in SMYD5. (**C**) Structural superposition of SET-I domains of SMYD5 and SMYD2. (**D**) Nearly buried cofactor binding sites.

**Figure 9 biomolecules-12-00783-f009:**
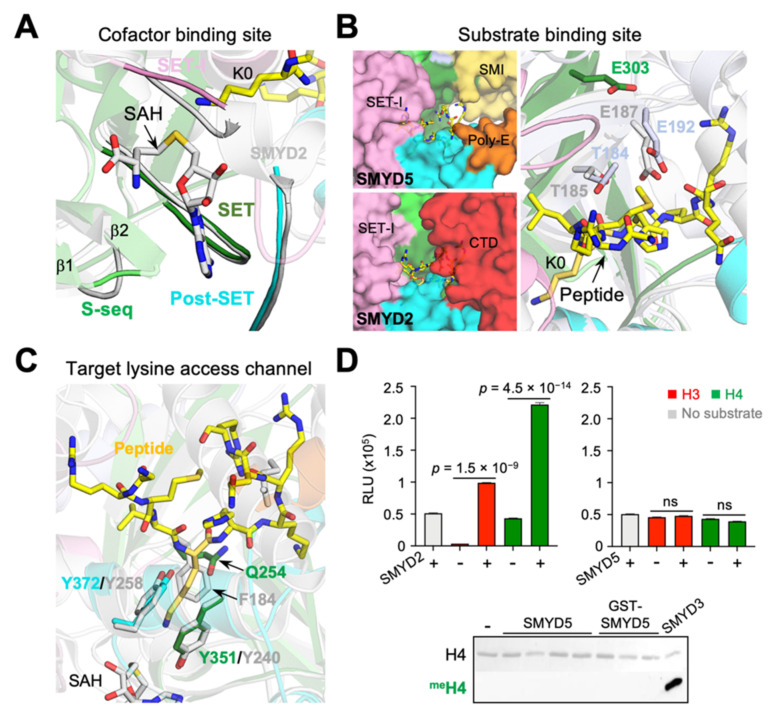
**Incomplete aromatic channel in the active site.** (**A**) Cofactor binding sites, (**B**) substrate binding sites, and (**C**) target lysine access channels of SMYD5 (colored) and SMYD2 (grey). (**D**) The methyltransferase activity of SMYD proteins on histones H3 and H4, assayed by the MTase-Glo methyltransferase assay (top), and by the antibody-based methyltransferase assay (bottom). ns, no significance.

## Data Availability

All data are contained in the manuscript and supporting information.

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
