# Peer review of "Unique SMYD5 Structure Revealed by AlphaFold Correlates with Its Functional Divergence"

_biomolecules, 2022, doi:10.3390/biom12060783_

Round 1
Reviewer 1 Report
Zhang and colleagues report in the manuscript their structural and functional analyses on mammalian SMYD5, one of the lysine methyltransferases lacking high-resolution structural details. They employed the recently developed modeling method of AlphaFold to investigate how the protein is folded and further revealed in vivo localization of SMYD5 directed by its N-terminal sequence that is distinct from what was believed previously. In the end, they propose a unique "crab"-like conformation of SMYD5 that is drastically different from those of all other SMYD proteins.
Overall, except for some points listed behind, their analyses are extensive, their results are convincing, and in my opinion their finding would be interesting to many readers in the relevant fields. Following are my few suggestions for them to consider when submitting the revision.
(1) There are two "poly-E" elements mentioned in the manuscript: one is the so-called "poly-E tract" at the C-terminus of SMYD5; the other is what they called "poly-E insertion" (line 347) or "poly-E loop" (Fig 8A, D). These two very similar terms can easily cause confusion to readers; I think one of them should be renamed, e.g. calling the insertion "NCL (negatively-charged loop)".
(2) All their structural analyses of different SMYD5 models were based on the precalculated models in the AlphaFold database, which only provides a single model for each protein. One should be very careful not to over-interpret such AlphaFold models, particularly for some regions with very low pLDDT values (per-residue confidence metric), which are called by the authors "AF score (%)" as seen in Fig 4B. This is exactly what concerns me in their conclusion that the "poly-E loop" completely shields the co-factor binding site of SMYD5 (Fig 8A, D). Notably, this so-called "poly-E loop", which spans residues ~195-215, is the least confident part in the core of the SMYD5 model (Fig 4B). Actually, this loop was predicted to adopt at least TWO distinct conformations when I ran the AlphaFold prediction myself, which generated 5 models in the end. My analyses showed that the two conformations are equally possible. However, unlike the one shown in their structural analyses, the other conformation contains a four turn helix packed against the all helical motif of the C-lobe and does NOT block the cofactor binding site at all! Therefore, I think it is too simple to say the cofactor binding site of SMYD5 is blocked by this loop. Instead, I think that this "poly-E loop" might function as a "gate" to regulate cofactor binding by alternating between the two conformations.
(3) Along the same line as in (2), the "poly-E loop" is both negatively charged and in close proximity to the "poly-E tract" at the C-terminus of SMYD5 (see Fig 7). It might be worth discussing how these two negatively charge regions might cooperate in regulate the in vivo function of SMYD5, e.g. DNA binding.
(4) A diagram of the AlphaFold models colored by pLDDT values would provide readers a direct view of the models for them to get a feeling the structural reliability of each residue. Similarly, the Predicted Aligned Error (PAE) diagram reported by AlphaFold, which provides distance between every residue pair and very useful to see long-distance and inter-domain cross-talks, would give a clear view how extensively the two lobes of SMYD5 interact with each other.
(5) In case the authors choose to run the AlphaFold prediction by themselves, which I think is necessary, a superposition cartoon plot of all five models generated for each SMYD5 would provide readers a clear view of structural variations (as mentioned above for the "poly-E loop") for the protein.
Author Response
Zhang and colleagues report in the manuscript their structural and functional analyses on mammalian SMYD5, one of the lysine methyltransferases lacking high-resolution structural details. They employed the recently developed modeling method of AlphaFold to investigate how the protein is folded and further revealed in vivo localization of SMYD5 directed by its N-terminal sequence that is distinct from what was believed previously. In the end, they propose a unique "crab"-like conformation of SMYD5 that is drastically different from those of all other SMYD proteins.
Overall, except for some points listed behind, their analyses are extensive, their results are convincing, and in my opinion their finding would be interesting to many readers in the relevant fields. Following are my few suggestions for them to consider when submitting the revision.
(1) There are two "poly-E" elements mentioned in the manuscript: one is the so-called "poly-E tract" at the C-terminus of SMYD5; the other is what they called "poly-E insertion" (line 347) or "poly-E loop" (Fig 8A, D). These two very similar terms can easily cause confusion to readers; I think one of them should be renamed, e.g. calling the insertion "NCL (negatively-charged loop)".
--- We renamed it “E-insertion” (Page 23, line 405). The labels in Fig 8 were revised accordingly.
(2) All their structural analyses of different SMYD5 models were based on the precalculated models in the AlphaFold database, which only provides a single model for each protein. One should be very careful not to over-interpret such AlphaFold models, particularly for some regions with very low pLDDT values (per-residue confidence metric), which are called by the authors "AF score (%)" as seen in Fig 4B. This is exactly what concerns me in their conclusion that the "poly-E loop" completely shields the co-factor binding site of SMYD5 (Fig 8A, D). Notably, this so-called "poly-E loop", which spans residues ~195-215, is the least confident part in the core of the SMYD5 model (Fig 4B). Actually, this loop was predicted to adopt at least TWO distinct conformations when I ran the AlphaFold prediction myself, which generated 5 models in the end. My analyses showed that the two conformations are equally possible. However, unlike the one shown in their structural analyses, the other conformation contains a four turn helix packed against the all helical motif of the C-lobe and does NOT block the cofactor binding site at all! Therefore, I think it is too simple to say the cofactor binding site of SMYD5 is blocked by this loop. Instead, I think that this "poly-E loop" might function as a "gate" to regulate cofactor binding by alternating between the two conformations.
---Thanks for your analysis and pointing out the flexible nature of this region. We did the AF prediction. Indeed, there is a helical conformation for this region in addition to the loop structure we described. We added discussion on the helical conformation and acknowledged that this region might be structurally flexible (Page 23, lines 406-415). Also, added a figure to show this conformation (Fig S10).
(3) Along the same line as in (2), the "poly-E loop" is both negatively charged and in close proximity to the "poly-E tract" at the C-terminus of SMYD5 (see Fig 7). It might be worth discussing how these two negatively charge regions might cooperate in regulate the in vivo function of SMYD5, e.g. DNA binding.
--- From the structure, it is not readily clear how these two parts could work together.
The poly-E tract is located near the substrate binding site and contributes to the formation of a deep substrate binding cleft. It might work together with SMI for regulating the binding of positively charged molecules, such as histones or protamines. This possibility had been discussed in the text. Our discussion on this possibility was based on the location and electrostatic properties of SMI. SMI is located across the cleft from the poly-E tract and has its negatively charged surface face the cleft, together with the poly-E tract shaping the substrate binding cleft.
However, the poly-E loop is located at the back side of the substrate binding cleft, so structurally, it is unlikely that it can contribute directly to the regulation of binding at the substrate binding cleft. One possibility is that the poly-E loop might indirectly do so by affecting the position of the poly-E tract through electrostatic repulsion. Usually, a spherical cutoff of 10 - 12 angstroms is applied to truncate the long-range electrostatic interactions. However, if we use this cutoff, there would be no electrostatic interaction between the poly-E tract and the poly-E loop, since the closest distances between these two regions in all SMYD5 structures we examined are more than 20 angstroms.
(4) A diagram of the AlphaFold models colored by pLDDT values would provide readers a direct view of the models for them to get a feeling the structural reliability of each residue. Similarly, the Predicted Aligned Error (PAE) diagram reported by AlphaFold, which provides distance between every residue pair and very useful to see long-distance and inter-domain cross-talks, would give a clear view how extensively the two lobes of SMYD5 interact with each other.
--- Added the figures showing the models colored by pLDDT values (Fig. S5). Added the PAE diagram for each model (Fig. S6)
(5) In case the authors choose to run the AlphaFold prediction by themselves, which I think is necessary, a superposition cartoon plot of all five models generated for each SMYD5 would provide readers a clear view of structural variations (as mentioned above for the "poly-E loop") for the protein.
--- We run AF prediction on each SMYD5, and added the superposition plots (Fig. S7)
Reviewer 2 Report
This paper is an interesting development, in that instead of presenting an experimentally determined structure, alpha-Fold predictions are presented. Within protein domains, alpha-Fold produces stunningly accurate results and here I fully trust the presented results. However, the relative orientation of protein domains and the conformation of low-confidence regions is not very well defined, which should be taken into account when describing the results, which, to my taste, should have been pointed out more clearly in the manuscript.
A very basic question, which came up by me when reading the manuscript: What is the function of SMYD5? No significant phenotype in SMYD5 knockout mice is mentioned. SMYD5 does not methylate histones H3 and H4 (see figure 9D + text) and no other potential substrate is mentioned. In the introduction, it is speculated, based on literature, that SMYD5 may be in volved in TLR4 signaling or maintaining chromosome integrity or in chromatin remodeling via protamine binding. Do these functions require histone methylation? Is SMYD5 able to methylate histones? Is a substrate known that is methylated by SMYD5?
The authors should either mention which substrate is methylated by SMYD5, or state that the substrate of SMYD5 is not known, and that it is not known whether SMYD5 has methyltransferase activity. Based on the structure analysis, they may then argue that SMYD5 likely has such an activity.
Not all abbreviations are explained in the text, e.g. I could not find explanation of SET and MYND.
Page 7, line 135: the overall difference between the two structures is defined by the root-mean-square difference between their Euclidian distances. “difference” is missing. In the methods section it is phrased correctly. One could also mention that it has been done for the Ca positions.
Page 11, figure 3C: what is SMYD5 and what is SMYD2?
Page 11, figure 3D, page 12, lines 182-183. Here a cleft of ~40 Å is mentioned. However, this distance is measured from the bottom of the cleft to the ill-defined poly-E tail, making this 40 Å basically a random number. The same is true for the width of the cleft, which, judging from figure 3D, has been measured diagonally and not perpendicular to the cleft, again making this a useless measure. (for this, I assume that the orientation, chosen in fig. 3D, is such that the cleft is shown at its widest).
Page 11, bottom: The poly-E tail does not have any interactions with the remainder of the protein and therefore, its predicted conformation is independent of the protein it is attached to. This is, as the authors mention on page 13, amongst the regions with the least confidence. We don’t know the conformation of this poly-E tail and it probably does not have a defined conformation.
Page 14, center part: Judging from figure 4, the depth of the cleft appear to depend on the orientation of the poly-E tail, which is ill-defined. I do not believe any discussion on differences of depth is warranted by the data presented, except that the dimensions of the cleft are flexible and may adapt to molecules of different sizes.
Page 15-17: the part of the nuclear localization signal is convincing.
Page 17, line 248: Also the first 7 residues are ill defined. For me, they are flexible and do not adopt any defined conformation. However, alpha-Fold predicts a different conformation for the different species, which for me, are just random conformations.
Page 20, lines278-280: I do not follow the way the authors come to a match percentage for the PXLXP motif of 68.4% per protein. They have matched 19 proteins, and 7 of those contain PXLXP motifs, which gives me a match percentage of 7/19=37% per protein. The authors have counted proteins with multiple PXLXP motifs multiple times in the numerator, but not in the denominator. With their method, if there would have been one protein with 19 PXLXP motifs and 18 proteins without such domains, they would have arrived at a 100% match.
Page 22-24: the prediction of the structure of the larger SET-I domain is convincing.
What I would like to see revised:
- some discussion on the substrate/lack of known substrate for SMYD5
- more cautious wording on low-confidence regions. One can safely assume, that these regions are disordered and only adopt a defined conformation when interacting with ligands or other proteins.
- Address the other points, mentioned in this review.
Author Response
This paper is an interesting development, in that instead of presenting an experimentally determined structure, alpha-Fold predictions are presented. Within protein domains, alpha-Fold produces stunningly accurate results and here I fully trust the presented results. However, the relative orientation of protein domains and the conformation of low-confidence regions is not very well defined, which should be taken into account when describing the results, which, to my taste, should have been pointed out more clearly in the manuscript.
A very basic question, which came up by me when reading the manuscript: What is the function of SMYD5? No significant phenotype in SMYD5 knockout mice is mentioned. SMYD5 does not methylate histones H3 and H4 (see figure 9D + text) and no other potential substrate is mentioned. In the introduction, it is speculated, based on literature, that SMYD5 may be in volved in TLR4 signaling or maintaining chromosome integrity or in chromatin remodeling via protamine binding. Do these functions require histone methylation? Is SMYD5 able to methylate histones? Is a substrate known that is methylated by SMYD5?
The authors should either mention which substrate is methylated by SMYD5, or state that the substrate of SMYD5 is not known, and that it is not known whether SMYD5 has methyltransferase activity. Based on the structure analysis, they may then argue that SMYD5 likely has such an activity.
--- That “SMYD5 was previously identified as a histone H4K20 methyltransferase” was mentioned in the original text (Page 25, line 486).
We agreed that this was not sufficient for readers to get an adequate background of SMYD5 functions.
We added further discussion on the currently known substrates of SMYD5 and their potential contributions to SMYD5 functions (Page 25, lines 487-491).
Not all abbreviations are explained in the text, e.g. I could not find explanation of SET and MYND.
--Added the full names of SET and MYND in the abstract (Page 1, line 11) and main text (Page 1, line 33).
Page 7, line 135: the overall difference between the two structures is defined by the root-mean-square difference between their Euclidian distances. “difference” is missing. In the methods section it is phrased correctly. One could also mention that it has been done for t.he Ca positions.
---Added “difference” (Page 7, line 142). Mentioned Ca atom used for calculation (Page 7, line 137).
Page 11, figure 3C: what is SMYD5 and what is SMYD2?
---Added the labels in Fig. 3C, and also added the explanations in the figure legends.
Page 11, figure 3D, page 12, lines 182-183. Here a cleft of ~40 Å is mentioned. However, this distance is measured from the bottom of the cleft to the ill-defined poly-E tail, making this 40 Å basically a random number. The same is true for the width of the cleft, which, judging from figure 3D, has been measured diagonally and not perpendicular to the cleft, again making this a useless measure. (for this, I assume that the orientation, chosen in fig. 3D, is such that the cleft is shown at its widest).
--- Good points for the poly-E tail. We re-measured the depth of the cleft using the rigid SMI domain as reference, which is ~35 angstroms. Both the text (Page 12, line 189) and figure were revised.
The width was measured perpendicularly. It appeared that our original labeling in the figure didn’t correctly reflect this. We have fixed the labeling. The width was also re-measured at several points using the rigid parts of the structure, which is ~ 25 angstroms, similar to our previous measurements.
Page 11, bottom: The poly-E tail does not have any interactions with the remainder of the protein and therefore, its predicted conformation is independent of the protein it is attached to. This is, as the authors mention on page 13, amongst the regions with the least confidence. We don’t know the conformation of this poly-E tail and it probably does not have a defined conformation.
--- We made a statement that the poly-E tail is flexible and might not have a defined conformation without binding to other molecules (Page 12, lines 207-208).
Page 14, center part: Judging from figure 4, the depth of the cleft appear to depend on the orientation of the poly-E tail, which is ill-defined. I do not believe any discussion on differences of depth is warranted by the data presented, except that the dimensions of the cleft are flexible and may adapt to molecules of different sizes.
--- We removed the overly discussed parts that were based on the flexible nature of the poly-E tail (Page 12). Removed Fig 4C and Fig 4D from the original figure.
Page 15-17: the part of the nuclear localization signal is convincing.
--- Thanks!
Page 17, line 248: Also the first 7 residues are ill defined. For me, they are flexible and do not adopt any defined conformation. However, alpha-Fold predicts a different conformation for the different species, which for me, are just random conformations.
--- We removed the sentence “can adopt at least four different conformations” and made a statement that the sequence might be “structurally disordered” (Page 16, line 302).
Page 20, lines278-280: I do not follow the way the authors come to a match percentage for the PXLXP motif of 68.4% per protein. They have matched 19 proteins, and 7 of those contain PXLXP motifs, which gives me a match percentage of 7/19=37% per protein. The authors have counted proteins with multiple PXLXP motifs multiple times in the numerator, but not in the denominator. With their method, if there would have been one protein with 19 PXLXP motifs and 18 proteins without such domains, they would have arrived at a 100% match.
--- The matches are the number of PXLXP motifs, not the number of proteins. Of interest is how many PXLXP motifs are present in a selected set of proteins. The match percentage can be more than 100% if the proteins in the selected set consist of, on average, more than one PXLXP motif. The same is for the background used for the enrichment analysis; the background is a random set of proteins: ~100,000 sequences or ~50,000,000 residues, where there are about 10,318 PXLXP matches (Methods).
We agreed that using the “percentage” and “%” appeared to be confusing. We changed “percentage” to “average” and changed “%” to the regular number in the text (Page 19, lines 333-334) and Methods (Page 33, lines 657-658).
Page 22-24: the prediction of the structure of the larger SET-I domain is convincing.
--- Thanks!
What I would like to see revised:
- some discussion on the substrate/lack of known substrate for SMYD5
--- Added discussion on the known SMYD5 substrates (Page 25, lines 487-491).
- more cautious wording on low-confidence regions. One can safely assume, that these regions are disordered and only adopt a defined conformation when interacting with ligands or other proteins.
--- We have made more cautious interpretation of the low-confidence regions, removed some overly discussed parts, and acknowledged that the low-confidence regions might be structurally disordered.
- Address the other points, mentioned in this review.
--- All other points were addressed.